# Protective Effect of *Limosilactobacillus fermentum* ME-3 against the Increase in Paracellular Permeability Induced by Chemotherapy or Inflammatory Conditions in Caco-2 Cell Models

**DOI:** 10.3390/ijms24076225

**Published:** 2023-03-25

**Authors:** Alex De Gregorio, Annalucia Serafino, Ewa Krystyna Krasnowska, Fabiana Superti, Maria Rosa Di Fazio, Maria Pia Fuggetta, Ivano Hammarberg Ferri, Carla Fiorentini

**Affiliations:** 1Institute of Translational Pharmacology, National Research Council of Italy (CNR), Via Fosso del Cavaliere 100, 00133 Rome, Italy; alex.degregorio@ift.cnr.it (A.D.G.); ewa.krasnowska@ift.cnr.it (E.K.K.); mariapia.fuggetta@ift.cnr.it (M.P.F.); 2National Centre for Innovative Technologies in Public Health, Istituto Superiore di Sanità, Viale Regina Elena, 299, 00161 Rome, Italy; fabiana.superti@iss.it; 3Association for Research on Integrative Oncology Therapies (ARTOI) Foundation, Via Ludovico Micara, 73, 00165 Rome, Italy; ivanoferri@yahoo.it (I.H.F.); carla.fiorentini@artoi.it (C.F.); 4SH Outpatient Oncology Clinic, Via dei Paceri 86/A, 47891 Falciano, San Marino; 5(IHF) Outpatient Oncology Clinic, Via dell’Indipendenza 20, 40121 Bologna, Italy

**Keywords:** dysbiosis, colon cancer, chemotherapy, *Limosilactobacillus fermentum* ME-3^®^, membrane permeability, tight junctions

## Abstract

Chemotherapy- or inflammation-induced increase in intestinal permeability represents a severe element in disease evolution in patients suffering from colorectal cancer and gut inflammatory conditions. Emerging data strongly support the gut microbiota’s role in preserving intestinal barrier integrity, whilst both chemotherapy and gut inflammation alter microbiota composition. Some probiotics might have a strong re-balancing effect on the gut microbiota, also positively affecting intestinal barrier integrity. In this study, we asked whether *Limosilactobacillus fermentum* ME-3 can prevent the intestinal paracellular permeability increase caused by the chemotherapeutic drug Irinotecan or by inflammatory stimuli, such as lipopolysaccharide (LPS). As an intestinal barrier model, we used a confluent and polarized Caco-2 cell monolayer and assessed the ME-3-induced effect on paracellular permeability by transepithelial electrical resistance (TEER) and fluorescent-dextran flux assays. The integrity of tight and adherens junctions was examined by confocal microscopy analysis. Transwell co-cultures of Caco-2 cells and U937-derived macrophages were used as models of LPS-induced intestinal inflammation to test the effect of ME-3 on release of the pro-inflammatory cytokines Tumor Necrosis Factor α, Interleukin-6, and Interleukin-8, was measured by ELISA. The results demonstrate that ME-3 prevents the IRI-induced increment in paracellular permeability, possibly by modulating the expression and localization of cell junction components. In addition, ME-3 inhibited both the increase in paracellular permeability and the release of pro-inflammatory cytokines in the co-culture model of LPS-induced inflammation. Our findings sustain the validity of *L. fermentum* ME-3 as a valuable therapeutic tool for preventing leaky gut syndrome, still currently without an available specific treatment.

## 1. Introduction

Colorectal cancer (CRC) is considered a global public health problem, the second leading cause of cancer-related death, and the third most commonly diagnosed cancer worldwide [1]. It is the result of the accumulation of genetic and/or epigenetic alterations that lead colonocytes to show uncontrolled hyperplasia and dysplasia [2]. Relevant roles are played by inflammation, dysbiosis, disruption of the gut barrier [3,4], and bacterial translocation [5], as they may lead to the development of CRC. The intestinal barrier plays a major role in health, and its dysregulation has been associated with a wide range of intestinal diseases, such as celiac disease, inflammatory bowel disease (IBD), and CRC [6,7,8,9,10]. The association of CRC with altered paracellular permeability of the gut epithelium and with bacterial translocation leads to a more efficient spread of metastasis by creating premetastatic niches or microenvironments that recruit metastatic cells in distant organs, such as the liver [11]. Consequently, paracellular permeability increase represents a severe element in patient disease evolution, accelerating the progression of CRC into metastatic disease and favoring a poor prognosis [12].

Chemotherapy, although necessary to treat CRC patients, alters the composition of microbiota (more specifically, the gut microbiota [13]), causes inflammation, damages tight junctions, and allows bacterial translocation [14]. Chemotherapy-dependent damage to the gut epithelium may contribute to the evolution of colon cancer, while the disease itself may worsen these specific side effects of chemotherapy. The increase in membrane permeability results from a plethora of events, one of the major ones being the functional alteration of an apical junctional complex made up of adherens junctions and tight junctions that provide strong connective bonds between epithelial cells [15]. Such cell–cell adhesion structures are essential to establish a barrier against free diffusion between different extracellular compartments of the body and to maintain homeostasis in various organ systems [16]. If altered or disrupted, the permeation of potentially harmful molecules and microorganisms from the intestinal lumen can result in a cascade of events, including immune activation and inflammation, eventually triggering the development of intestinal and systemic diseases, including cancer [17]. So far, no efficacious solution has been proposed to counteract augmented intestinal permeability, whatever the cause.

As stated above, leaky gut syndrome influences microbiota composition, which, if impaired, may be associated with several health consequences. Dysregulated microbiota could be treated with probiotics with health-promoting effects capable of stabilizing the microbial communities, thus gaining an essential position in medical health care. In this context, *Limosilactobacillus fermentum* ME-3^®^ (according to the nomenclature updated in 2020 [18]; formerly, *Lactobacillus fermentum*), a patented strain of Lactobacillus species [19], may represent a suitable candidate for investigation, having antimicrobial, antioxidant, and antiatherogenic properties and expressing several beneficial properties [19,20,21,22].

The general purpose of this study was to investigate the possibility of counteracting the alteration of membrane permeability that occurs in the intestinal epithelium in the presence of colon cancer. More specifically, in an intestinal cell line derived from human colon cancer—Caco-2 cells—we verified whether *L. fermentum* ME-3^®^ could prevent or repair the intestinal membrane damage caused by chemotherapy and prevent or downregulate LPS-induced inflammation. The results obtained clearly demonstrate that preincubation of Caco-2 cells with ME-3^®^ prevents the increase in paracellular permeability induced by chemotherapy, most probably by modulating the expression and localization of tight junction proteins. In addition, *L. fermentum* ME-3^®^ prevents the increments in membrane permeability and pro-inflammatory cytokine production in LPS-induced inflammation in Caco-2 cells.

## 2. Results

### 2.1. L. fermentum ME-3^®^ Does Not Alter the Effects of Irinotecan on the Proliferation and Vitality of Caco-2 Cells

The starting point was to verify whether *L. fermentum* ME-3^®^ (ME-3^®^) impacts the viability of colon cancer cells and, more importantly, whether it can inhibit the cytotoxic effect of chemotherapy. For this study, we used Irinotecan (IRI), a derivative of the cytotoxic alkaloid camptothecin, approved by the Food and Drug Administration in 1996 for the treatment of advanced colorectal cancer but currently used, alone or in combination with other antitumor drugs, to manage and treat a variety of other solid tumors [23]. IRI is one of the main topoisomerase I inhibitors, possessing anticancer efficacy by different mechanisms, including the formation of a ternary Irinotecan-topoisomerase I-nicked DNA complex that leads to the inhibition of DNA replication and transcription, which ultimately results in apoptotic cell death [24]. The kinetics of Caco-2 cell viability after treatment with increasing concentrations of IRI over time periods ranging from 24 to 72 h are illustrated in Figure 1A. Our results show that IRI significantly reduced the exponential growth of cancer cells in a concentration- and time-dependent manner. The half-maximal inhibitory concentration (IC50) of treatment with IRI for 24 h (155 µg/mL) was used for all subsequent experiments.

In order to evaluate whether ME-3^®^ impacts the vitality of colon cancer cells, Caco-2 cells were incubated for 24 h in the presence of a suspension of two different concentrations of ME-3^®^ (10^6^–10^7^ colony forming units (CFUs)/mL). The viability assay showed that this probiotic strain had no toxic effects on Caco-2 cells (Figure 1B, left panel). More importantly, ME-3^®^ did not counteract the chemotherapy-induced cytotoxicity, which was preserved even when cancer cells were preincubated with ME-3^®^ before IRI treatment. Indeed, as shown in the right panel of Figure 1B, ME-3^®^-pretreated Caco-2 cells were no more viable than those treated with IRI alone (31 vs. 35%). This was also confirmed by the results of confocal laser scanning microscopy (CLSM) observations after nuclei staining with Hoechst (Figure 1C), which revealed that the IRI-induced increment in the number of cells that exhibited fragmented and/or condensed nuclei, indicative of apoptosis induction, was also observed when the cultures were pretreated with ME-3^®^. Taken together, these results indicate that *L. fermentum* ME-3^®^ does not alter Caco-2 cell viability and does not inhibit and/or attenuate the cytotoxic effects of IRI.

### 2.2. L. fermentum ME-3^®^ Protects Caco-2 Monolayers from the Increase in Membrane Permeability Caused by Irinotecan Treatment

Preventing leaky gut is of primary importance as it is a mechanism of colon cancer progression that accelerates metastatic spreading [6]. Changes in the transepithelial electrical resistance (TEER) and paracellular permeability of polarized Caco-2 monolayer cells (see Model **a** described in the MM section) are considered specific and sensitive biomarkers of intestinal barrier integrity and function. Indeed, TEER gradually increased over time as a result of cell polarization and junctional differentiation (Appendix A).

As depicted in Figure 2A, after 24 h, TEER values of cultures treated with IRI significantly decreased by 14.2% compared with the untreated control, confirming that this in vitro model of the intestinal barrier is a suitable cellular system for evaluating the chemotherapy-induced damage of paracellular permeability. In order to assess the ability of ME-3^®^ to prevent this barrier dysfunction, the polarized monolayer cells were per-incubated with the ME-3^®^ suspension for 24 h, and TEER values were then measured after an additional 24 h of IRI treatment (Figure 2A). Interestingly, the reduction in TEER values recorded with IRI treatment was counteracted by ME-3^®^ pretreatment. No difference in transmembrane electrical impedance was recorded after ME-3^®^ incubation alone, thus confirming the safety of this probiotic in Caco-2 cells, in accordance with the cell viability results.

The ability of ME-3^®^ to prevent IRI-induced barrier dysfunction was also investigated by measuring the paracellular flux of FITC-dextran fluorescent tracers across the Caco-2 cell monolayer. As shown in Figure 2B, treatment with IRI rapidly increased the unidirectional flux of FITC-dextran from the apical to the basolateral chamber. This increase, which was recorded after only two hours, became statistically significant after 24 h of IRI treatment. When the Caco-2 cell monolayer was pretreated with ME-3^®^, the alteration of paracellular permeability induced by IRI was significantly counteracted (Figure 2B), in line with the results obtained by the TEER assay (Figure 2A), thus confirming that ME-3^®^ was able to protect our gut barrier model by IRI-induced stress on the cell-to-cell adhesion.

### 2.3. L. fermentum ME-3^®^ Protects Caco-2 Cell Monolayers from the IRI-Induced Alteration of Tight Junction Integrity

Tight junctions (TJs) consist of transmembrane proteins, including occludins and claudins, and are considered the principal determinants of mucosal permeability [25]. Confocal microscopy observation (Figure 3A) and quantitative analysis of Occludin and Claudin-3 colocalization indexes (CIs; Figure 3B) showed that, upon exposure to 155 µg/mL (IC50) IRI for 24 h, the differentiated Caco-2 cell monolayer exhibited decreased TJ integrity compared with the untreated control, as demonstrated by the significantly (*p* < 0.05) reduced expression and colocalization (approximately 1.8-fold decrements vs. the untreated control) of these two TJ components at the cell–cell contacts.

Conversely, pretreatment with ME-3^®^ before IRI addition to the culture medium significantly (*p* < 0.05) prevented the IRI-induced TJ disruption, as indicated by the Occludin/Claudin-3 colocalization indexe which was similar to that recorded for the untreated control (approximately 1.5-fold increments vs. the IRI-treated cells). No significant effects on TJ integrity were recorded under ME-3^®^ treatment alone, in line with the results obtained by TEER and FITC-dextran flux assays (Figure 2).

### 2.4. L. fermentum ME-3^®^ Counteracts the IRI-Induced Alteration of Expression and Localization of Proteins Controlling the Adherens Junctions

The assembly of the TJs is facilitated by the formation of the adherens junctions (AJs), which provide strong connective bonds between epithelial cells and consist of adhesion molecule complexes made up of protein families comprising cadherins and catenins [26].

On this basis, to provide additional evidence for the efficacy of ME-3^®^ in preserving intestinal barrier integrity, we also explored the effects of this probiotic strain on E-cadherin and β-catenin expression and distribution at the cell–cell contacts. As shown in Figure 4, IRI treatment reduced the expression of both of these AJ components at the cell membranes of the differentiated Caco-2 cell monolayer (approximately 1.3-fold reductions vs. the untreated control for both proteins), even if the decrement recorded in the quantitative analysis of fluorescence intensity performed through confocal microscopy (Figure 4B) was significant (*p* < 0.01) only for β-catenin. Pretreatment with ME-3^®^, performed 24 h before IRI addition to the culture medium, significantly (*p* < 0.05) prevented the IRI-induced reduction in both AJ components, leading to expression values of β-catenin and E-cadherin at the cell-to-cell contacts significantly higher than those recorded in the untreated control (approximately 2-fold and 1.2-fold increments vs. the control, respectively; Figure 4B). Similar to what has been observed for TJ integrity, no significant effects on AJ components were recorded under ME-3^®^ treatment alone.

### 2.5. L. fermentum ME-3^®^ Prevents the Increases in Membrane Permeability and Pro-Inflammatory Cytokine Release Caused by LPS-Stimulated Inflammation in Caco-2 Cells

To verify whether ME-3^®^ could also be protective when an inflammatory condition induces increase in paracellular permeability, we employed an in vitro model of intestinal inflammation consisting of co-cultures of differentiated human histiocytic lymphoma cells (dU937) and Caco-2 cells subjected to an inflammatory stimulus with LPS (see Model **b** described in the MM section). The co-culture of Caco-2 with dU937 did not alter the cell–cell contacts, as demonstrated by the quite similar measures of dextran flux across the Caco-2 monolayer cells performed at T0 in Models **a** and **b** (FITC-dextran intensity at T0 was equal to 2113 and 2141.5 in Models **a** and **b**, respectively). As reported in Figure 5A, the FITC-dextran flux across the Caco-2 cell monolayer rapidly and significantly increased with incubation time in LPS-inflamed co-cultured dU937 cells, indicating that the paracellular permeability of the Caco-2 monolayer was impaired. However, when the monolayer was pretreated with ME-3^®^, the FITC-dextran flux was significantly lower than that recorded under LPS insult, confirming that ME-3^®^ is also able to prevent the damage to intestinal barrier integrity induced by inflammation.

Given that inflammation produces a large number of inflammatory mediators, we also assessed whether ME-3^®^ pretreatment influenced the LPS-induced release of some pro-inflammatory cytokines in the co-culture model. The results reported in Figure 5B show that in the apical chamber of the co-cultures, LPS did not affect IL-6 release, while it significantly increased the levels of TNF-α (13.09 pg/mL) and IL-8 (33.74 pg/mL) produced by Caco-2 cells. Moreover, in the same experimental condition, TNF-α (505.95 pg/mL) and IL-6 (21.83 pg/mL), but not IL-8, produced by dU937 cells in the basolateral chamber were significantly increased after treatment with LPS. When dU937 cells were pretreated with ME-3^®^ (ME-3^®^ does not affect dU937 cell viability), the production of TNF-α by Caco-2 cells in the apical chamber was not significantly inhibited; still, it was considerably reduced, from 505.95 to 129.63 pg/mL, in the basolateral chamber. The IL-6 release by dU937 cells in the basolateral compartment, but not that by Caco-2 cells in the apical one, was also significantly decreased, from 21.83 to 11.08 pg/mL, while the apical but not the basolateral IL-8 production was significantly inhibited by ME-3^®^, decreasing from 33.74 to 18.26. These results demonstrate that ME-3^®^ has anti-inflammatory potential by reducing the release of cytokines, such as TNF-α, IL-6, and IL-8, in a co-culture model of intestinal inflammation.

## 3. Discussion

The principal outcome of the present study is that *Lactobacillus fermentum* ME-3^®^ can prevent the increase in intestinal permeability caused by chemotherapy against colorectal cancer. Of importance, the preventive action of the probiotic was also effective when the membrane damage was triggered by LPS-induced inflammation, thus evidencing that the protection offered by ME-3^®^ is not strictly dependent on the insult. These results potentially represent a breakthrough in medicine, as there are no specific therapies to counteract augmented intestinal permeability, irrespective of the inducing factor.

As stated in the Introduction, colorectal cancer represents a significant health problem, being one of the most common forms of cancer, with a high mortality rate leading to poor long-term survival [1,27]. Chemotherapy, although a fundamental step in therapy for cancer patients, causes significant gut toxicity, leading to several consequences and side effects, some of which remain difficult to control [11,28,29]. In recent years, several studies have focused on the roles played by a dysfunctional gut barrier and altered microbiota composition on inflammation and colon cancer development [3,4]. The gut microbiota of patients with cancer and undergoing chemotherapy is altered [13], and, in this context, it is emerging that some probiotics, if properly used, can have strong re-balancing effects on the gut microbiota and therefore positively impact some of the adverse events that occur in cancer [30]. Furthermore, new evidence supports the hypothesis that using some selected probiotics may represent a feasible approach to effectively protect patients against possible severe consequences of chemotherapy [31,32]. Our investigation explored the ability of *L. fermentum* ME-3^®^ to prevent the membrane damage caused by colon-cancer-related therapy and LPS-induced inflammation, both of which can increase patient discomfort and cause side effects, in addition to contributing to systemic inflammation and the spread of bacterial products. *Lactobacillus fermentum* ME-3^®^ is a probiotic strain of human origin with health-promoting characteristics patented by Tartu University in 2006 [19]. ME-3^®^ was chosen since its administration has been proven to increase gut microbiota diversity, thus diminishing the risk of gut diseases [33], reducing lipid peroxidation [20,22], and raising serum levels of glutathione [34,35,36] and of the antioxidant enzyme paraoxonase 1 (PON1) [22]. It is worth noting that oxidative stress, PON1, inflammation, and cancer development are strictly interconnected [37].

As a first step, we evaluated the effects of *Lactobacillus fermentum* ME-3^®^ on Caco-2 cell monolayers and found that it neither affected cell viability nor inhibited the cytotoxic effect of Irinotecan. These data are relevant since they demonstrate that ME-3^®^ preincubation does not nullify the toxicity of chemotherapy in cancer cells. Then, to assess cell monolayer permeability, we used two independent approaches, namely, TEER and FITC-dextran paracellular flux assays. These two methodologies are related to two different cellular parameters that are regulated independently [38]. Hence, their simultaneous employment reinforces the value of the data we obtained via membrane permeability measurement. We showed that ME-3^®^ could prevent the increase in paracellular permeability induced by Irinotecan, proving that ME-3^®^ protects against cell barrier damage. Irinotecan has been shown to disrupt tight junction proteins and cause severe histological damage in the intestines, consequently leading to mucosal barrier dysfunction [39]. When healthy, the epithelial barrier is impermeable to toxins, pathogens, and antigens while maintaining a selective permeability via the paracellular and transcellular pathways [16]. In particular, the paracellular pathways, which control the passage of molecules in the spaces between adjacent cells, are regulated by an apical junctional complex made up of adherens and tight junctions. If these junctions are disrupted, the permeation of potentially harmful molecules and organisms from the intestinal lumen can induce immune activation and inflammation, eventually triggering the development of intestinal and systemic diseases, including cancer [17]. Our results show that *L. fermentum* ME-3^®^ protects cultured cancer cells against the harmful effects of Irinotecan on the apical junctional complex, mainly consisting of decreased expression and altered localization of proteins controlling tight and adherens junctions. This is particularly relevant since, while battling cancer cells, Irinotecan damages tight junctions and induces bacterial translocation [14], this last being involved in the carcinogenesis of gut cancers and their prognosis [5].

Of great importance, besides Irinotecan, we demonstrated that this probiotic could also protect against the membrane damage provoked by LPS-induced inflammation, LPS excess being a prevalent trait of dysbiosis, chronic gut inflammation, and colon cancer [40,41,42]. To address this point, a Caco-2/dU937 co-culture model of intestinal inflammation was treated with LPS, thus provoking FITC-dextran flux across the Caco-2 monolayer. Preincubation with ME-3^®^ significantly reduced the flux, showing that it can prevent damage to the intestinal barrier caused by inflammation. Furthermore, ME-3^®^ was proved to inhibit the release of TNF-α, IL-6, and IL-8 from both sides of the co-culture model. IL-6 is a pleiotropic cytokine with anti-inflammatory, pro-inflammatory, or immunosuppressive functions, although its overexpression in the gut can cause an increase in TJ permeability [43]. IL-8 is a well-known pro-inflammatory cytokine involved in many pathways which has the ability to increase gut permeability by downregulation of TJ in a dose- and time-dependent manner [44]. Tumor necrosis factor-α (TNF-α), one of the principal effectors of IBD inflammation, may induce alterations in gut permeability by modulating TJ protein transcription, while its antagonists (anti-TNF-α) can improve intestinal permeability [45]. However, TNF-α also leads to altered permeability, inducing apoptosis of enterocytes, increasing their rate of shedding and hindering the redistribution of TJs that should seal the gaps left. Our data strongly indicate that *L. fermentum* ME-3^®^ possesses anti-inflammatory potential, preventing the LPS-induced increase in membrane permeability and inhibiting the release of pro-inflammatory cytokines in a model of intestinal inflammation.

Regarding the mechanistic aspects of the protective effects of ME-3 on the intestinal barrier, we did not deepen understanding of these, nor have we have available relevant experimental data, being able to only make some hypotheses based on data reported in the literature. In our view, the primary mechanism that *L. fermentum* ME-3 uses to maintain intestinal integrity might be the upregulated expression and distribution of proteins belonging to the apical junctional complex, as already reported for other probiotics [46]. The reinforcement of the mucosal barrier can halt the damage caused by Irinotecan and LPS-induced inflammation. In addition, the anti-inflammatory properties exhibited by ME-3 that prevent the secretion of pro-inflammatory cytokines are also capable of enhancing barrier function. Furthermore, an additional hypothesis is that the strong antioxidative properties of *L. fermentum* Me-3 [19] permit a protective effect against Irinotecan by reverting the oxidative imbalance evoked by chemotherapy. A similar mechanism was observed in CaCo-2 incubated with Irinotecan and the flavonoid luteolin, which can attenuate irinotecan-induced oxidative stress by its scavenging property [47].

Even if our work possesses the intrinsic limitation of an in vitro study and the data obtained should be subsequently validated by studies on animals and/or human subjects, taken altogether, these results represent a step forward, since gut permeability and the consequent triggering of immune dysfunction are major problems in modern society. The use of antibiotics, pesticides, and many more chemicals, as well as chemotherapy, may cause gut microbiota dysbiosis, local and systemic inflammation, the passage of biochemical products, including toxins from bacteria in the bloodstream, and bacterial translocation [5]. These events contribute to developing chronic gut inflammation and disease, autoimmune disease, chronic degenerative disease, colorectal cancer, and potentially even cancer progression. In the case of colon cancer, all the above-mentioned dysfunctional effects are allies of the disease, and the known damage to the gut epithelium caused by Irinotecan treatment creates additional trouble for the patient. Up to now, there is no established pharmacological cure for gut permeability, although research in the field is intensively ongoing. The search for therapies that may prevent gut damage in the treatment of colon cancer is of primary importance; we believe that it may contribute to various aspects ranging from the life quality of patients to possible therapy outcomes and, potentially, even prognosis.

In conclusion, *Lactobacillus fermentum* ME-3^®^ has been demonstrated to prevent gut barrier damage, protect TJ and AJ expression and distribution, and reduce the release of pro-inflammatory cytokines in human colon adenocarcinoma cultures. Based on our results, colon cancer patients undergoing chemotherapy with Irinotecan could benefit from ME-3^®^ properties, both in terms of improved gut health and reduced local inflammation by modulation of their microbiota. Further research is needed to exploit all the potentialities of *Lactobacillus fermentum* ME-3^®^ and more clearly define its mode of action.

## 4. Materials and Methods

### 4.1. Reagents and Chemicals

*L. fermentum* ME-3^®^ (cod. 643749-10B-1; BIODIS, 92532 Levallois-Perret, France) was resuspended in antibiotic-free DMEM at the established concentration, expressed as number of CFUs (colony forming units). Irinotecan (Irinotecan hydrochloride, cod. I1406; Sigma Aldrich, St. Louis, MO, USA) was purchased from Merck (KGaA, Darmstadt, Germany).

### 4.2. Cell Cultures and Cellular Models

The human colon cancer cells (Caco-2) and the human histiocytic lymphoma cells (U937) were obtained from the American Type Culture Collection (Manassas, VA, USA). Caco-2 cells were grown in Dulbecco’s Modified Eagle Medium (DMEM), and U937 cells were cultivated in RPMI 1640 medium. Both media, hereafter called complete media (CM), were supplemented with inactivated 10% fetal bovine serum (FBS), penicillin (100 IU/mL), and streptomycin (100 μg/mL) (Invitrogen, Carlsbad, CA, USA). The cells were cultured in a humidified incubator with 5% CO_2_ and kept at 37°C. All the treatments with ME-3^®^ were performed in complete media without antibiotics.

The following cellular models were used (schematized in Figure 6):

**Model a**. For the establishment of an in vitro epithelial barrier model, Caco-2 cells (0.1 × 10^6^ cells/well) were seeded into the apical chamber of 24-well transwell inserts, 3.0 μm pore size, (Corning, New York, NY, USA). The cells were grown and differentiated over 21 days. The CM (500 µL in the apical chamber and 500 μL in the basolateral chamber) was refreshed every other day for the first two weeks and then every day until the cells were fully differentiated (week three).

**Model b**. For the establishment of an in vitro model of intestinal inflammation, a Caco-2/U937 co-culture was performed. Caco-2 cells (0.1 × 10^6^ cells/well) were seeded, as previously described for Model a, in the apical chamber of a 24-well transwell and cultured for 21 days. U937 cells (2 × 10^5^ cells/well) were stimulated with 50 ng/mL of phorbol-12-myristate-13-acetate (PMA; Sigma-Aldrich, St. Louis, MO, USA) to differentiate into macrophage-like cells in a 24-well plate for 24 h, as described in the literature [48]. Then, the differentiated U937 macrophages (dU937) were stimulated with 1 µg/mL of LPS (Sigma-Aldrich) for 4 h and co-cultured with Caco-2 monolayer cells (week three) for 24 h.

### 4.3. Cytotoxicity Test

The cytotoxic effects of IRI in Caco-2 cells were assessed by MTT [3-(4,5-dimethylthiazol-2-yl)-2,5-diphenyltetrazolium bromide] assay. Caco-2 cells (2.5 × 10^3^ cells/well) were seeded into 96-well plates, and after overnight incubation the medium was replaced with CM containing increasing concentrations of IRI (12.5, 25, 50, 100, and 200 μg/mL). The control group was cultured using CM alone. After 24, 48, or 72 h of treatment, 1 mg/mL of MTT solution (Sigma-Aldrich) was added, and samples were incubated for 4 h. A quantity of 100 μL of DMSO was then added, and formazan production was detected by determining the optical density (OD) at 570 nm using an ELISA reader (Multiskan EX, Thermo Fisher Scientific, Waltham, MA, USA). From the dose–response curve thus obtained, a 50 inhibitory concentration (IC50) was determined as an index of antitumor activity, and the IC50 of 24 h was used for all treatments.

The effect of ME-3^®^ on Caco-2 cell viability was assessed by Trypan blue (TB) dye exclusion assay. Briefly, Caco-2 cells (0.1 × 10^6^ cells/well) were seeded into a 24-well plate and after overnight incubation were treated with ME-3^®^ (10^6^ CFU/mL and 10^7^ CFU/mL) for 24 h. Then, cells were detached using trypsin solution 0.05% and manually counted in TB (Corning, Glendale, AZ, USA).

### 4.4. Permeability of Intestinal Barrier Evaluation

The effect of IRI and ME-3^®^, alone or in combination, on the membrane integrity of Caco-2 monolayer cells was evaluated using Models **a** and **b**, described above.

The apical and basolateral chambers of the Caco-2 monolayer cells in Model **a** were treated with IRI (155 µg/mL, i.e., IC50) and ME-3^®^ (10^7^ CFU/mL) alone or in combination (as pretreatment with ME-3^®^ for 24 h and IRI for a further 24 h).

The apical and basolateral chambers of the Caco-2 monolayer cells in Model **b** were treated with ME-3^®^ (10^7^ CFU/mL) for 24 h and then co-cultured with dU937 stimulated with LPS (1 µg/mL). The control was represented by dU937 not stimulated with LPS.

The cell-to-cell contact integrity was evaluated by transepithelial electrical resistance (TEER) and measurement of paracellular permeability.

### 4.5. Transepithelial Electrical Resistance (TEER) Measurement

TEER was used as a measure of cell monolayer integrity and was assessed before (t0) and after (t24) all treatments. TEER was measured based on previously described approaches [49,50]. Briefly, the TEER of the Caco-2 monolayer cells in Model **a** was determined using a Millicell^®^-ERS-2 V-Ohm meter (Millipore, Burlington, MA, USA). The electrode was immersed at a 90° angle, with one tip in the basolateral chamber and the other in the apical chamber (without touching the cells). Three transwells were tested for each condition, and an insert without cells was used as a blank; its mean resistance was subtracted from all samples. We calculated the TEER value according to the following formula:TEER = (TEERs − TEERb) × A
where TEERs is the TEER value (Ω) of the measured transwell chamber with cells, TEERb is the TEER value (Ω) of the transwell chamber without cells, and A is the membrane area of the transwell cavity (0.33 cm^2^). Caco-2 monolayers with TEER values of >300 Ω∙cm^2^ were used in this study (see Appendix A).

To reduce the error due to the variability in the values measured at t0 (baseline), a correction factor was introduced which normalized the t24 of each well with respect to its own t0, and the values were expressed as TEER % of the baseline according to the following formula:(TEERt t24/TEERc t24 × TEER c t0/TEERt t24) × 100
where TEERt represents the TEER value (Ω) of the measured transwell chamber with treated cells and TEERc is the TEER value (Ω) of the transwell chamber in the control group.

### 4.6. Paracellular Permeability Measurement by FITC-Dextran Flux Assay

In addition to TEER measurement, the paracellular flux of tracers across the cell monolayer also reflects the permeability of the intestinal barrier. The epithelial permeability of Caco-2 monolayer cells in Models **a** and **b** was assessed by measuring the tracer flux from the apical to the basolateral chamber. The treatment was performed in medium containing 1 mg/mL of fluorescein isothiocyanate (FITC)-dextran (FITC-4kDa dextran; Sigma-Aldrich), which was added to the apical chamber. The fluorescence intensity in the basolateral chamber was measured with an automated ELISA reader (a VICTOR 3™ Multilabel Plate Reader obtained from PerkinElmer, Waltham, MA, USA) at different time points (2, 4, 6, and 24 h). The excitation and emission wavelengths for FITC-dextran were 490 and 520 nm, respectively.

### 4.7. Immunofluorescence Analyses and Confocal Laser Scanning Microscopy (CLSM)

Immunofluorescence analysis was carried out on Caco-2 monolayer cells grown as described in Model a. Cells were fixed with 4% paraformaldehyde (Sigma-Aldrich, St. Louis, MO, USA) and permeabilized with 0.2% Triton X-100 (Sigma-Aldrich). β-Catenin and E-cadherin single immunostaining was performed using the following antibodies: the mouse monoclonal antibodies against β-catenin (cod. 610154; BD Transduction Labs, Palo Alto, CA, USA. 1:250) or the mouse monoclonal antibodies against E-cadherin (cod. 610182; BD Transduction Labs, 1:50). For Occludin/Claudin-3 double immunostaining, the specific mouse monoclonal antibodies against Occludin (cod. 33-1500; Invitrogen, Waltham, MA, USA; 1 µg/mL) and the rabbit polyclonal antibody Claudin-3 (cod. 34-1700; Invitrogen, 1:100) were used. Primary antibodies were revealed with Alexa Fluor 488-conjugated or TRITC-conjugated anti-mouse or anti-rabbit IgG (Molecular Probes, Eugene, OR, USA), as appropriate. Cell nuclei were counterstained with 0.2% Hoechst (Sigma-Aldrich). Samples were observed with a confocal microscope (LEICA TCS SP5; Leica Instruments, Heidelberg, Germany). Quantification of immunofluorescence staining of the AJ components β-catenin and E-cadherin and of the colocalization of Occludin/Claudin-3 TJ components (Pearson’s correlations (PCs) and colocalization rates (CRs)) was obtained using the Leica application suite for advanced fluorescence software (Leica Instruments, Heidelberg, Germany); the mean fluorescence intensity and the mean PCs and CRs were calculated by analyzing a minimum of 10 fields/sample in a blind fashion. The colocalization index (CI) was obtained using the formula: CI = CR × PC.

### 4.8. Determination of Cytokines in the Intestinal Inflammation Model

The effect of ME-3^®^ on inflammatory cytokine release, including TNF-α (ELISA kit; BD Bioscience, Franklin_Lakes, NJ, USA), IL-6, and IL-8 (ELISA kit; Mabtech, Sigma-Aldrich), was evaluated using the Caco-2/U937 co-culture described above in Model **b**. In this case, dU937 cells were pretreated with ME-3^®^ and then stimulated with LPS.

The cytokines released in the medium from the apical or basolateral chamber were detected using the ELISA kits according to the manufacturer’s instructions. The control group was represented by dU937 not stimulated with LPS.

### 4.9. Statistical Analysis

All the experiments were performed three times, and the data were expressed as the means + SDs. Statistical analysis was conducted using GraphPad Prism software version 5.01. Significant differences amongst samples were assessed using the two-tailed Student’s *t*-test or one-way ANOVA with Tukey’s test. Differences were considered statistically significant when the *p*-value was <0.05.

## 5. Conclusions

The obtained results can represent a significant step beyond the current state of the art, since leaky gut syndrome affects a very high percentage of the population that frequently uses antibiotics rather than chemotherapy. Plus, it is a frequent problem in gluten-intolerant individuals, as well as in those suffering from chronic intestinal dysbiosis. Hence, given its efficacy, the applicability of *L. fermentum* ME-3^®^ is potentially enormous and broad-spectrum. These findings open new therapeutic horizons at a historical moment in which the microbiota is being recognized as a useful therapeutic tool for various pathologies.

## Figures and Tables

**Figure 1 ijms-24-06225-f001:**
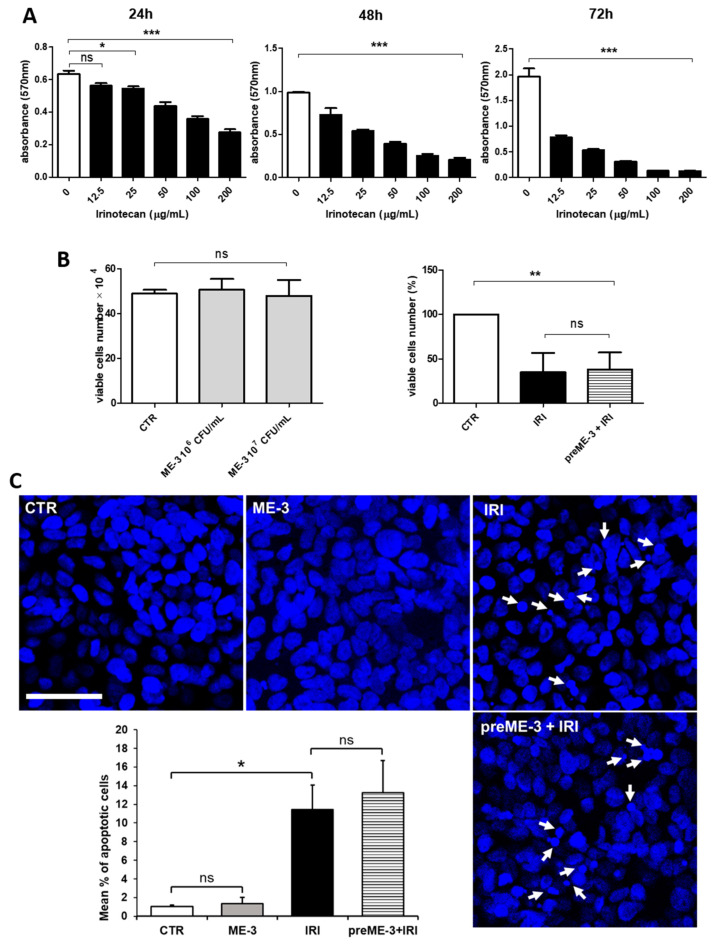
Effects of Irinotecan and ME-3^®^ on the proliferation and vitality of Caco-2 cells. (**A**) Time course of the antiproliferative effect of IRI on the Caco-2 cell line. Cell proliferation was determined by MTT assay. The values are expressed as the means of three experiments ± SDs. (**B**) Viability of Caco-2 cells treated with two different concentrations of ME-3^®^ for 24 h (left) or pretreated with ME-3^®^ (10^7^ CFU/mL) for 24 h and then with IRI (155 µg/mL) for an additional 24 h (i.e., preME-3^®^ + IRI) (right). Cell proliferation was determined by cell counts in trypan blue. Values are expressed as the means of three separate counts ± SDs. Asterisks in panels (**A**,**B**) represent significant differences between indicated groups: * *p* < 0.05, ** *p* < 0.01, *** *p* < 0.001; ns = not significant. (**C**) Representative images obtained by confocal laser scanning microscopy and quantitative analysis (bar graph) of the percentage of Caco-2 cells exhibiting fragmented and/or condensed nuclei indicative of apoptosis (white arrows) in controls (untreated and ME-3^®^ treated cells) and in IRI-treated cultures grown in the presence or absence of ME-3^®^ pretreatment. Bar = 50 μm. The quantitative analysis of apoptotic cells, performed using ImageJ processing software, was carried out by analyzing a minimum of 800 cells/sample, and the results are the means ± SDs from three independent experiments (*n* = 3). Significance (two-tailed Student’s *t*-test): * *p* < 0.05; ns = not significant.

**Figure 2 ijms-24-06225-f002:**
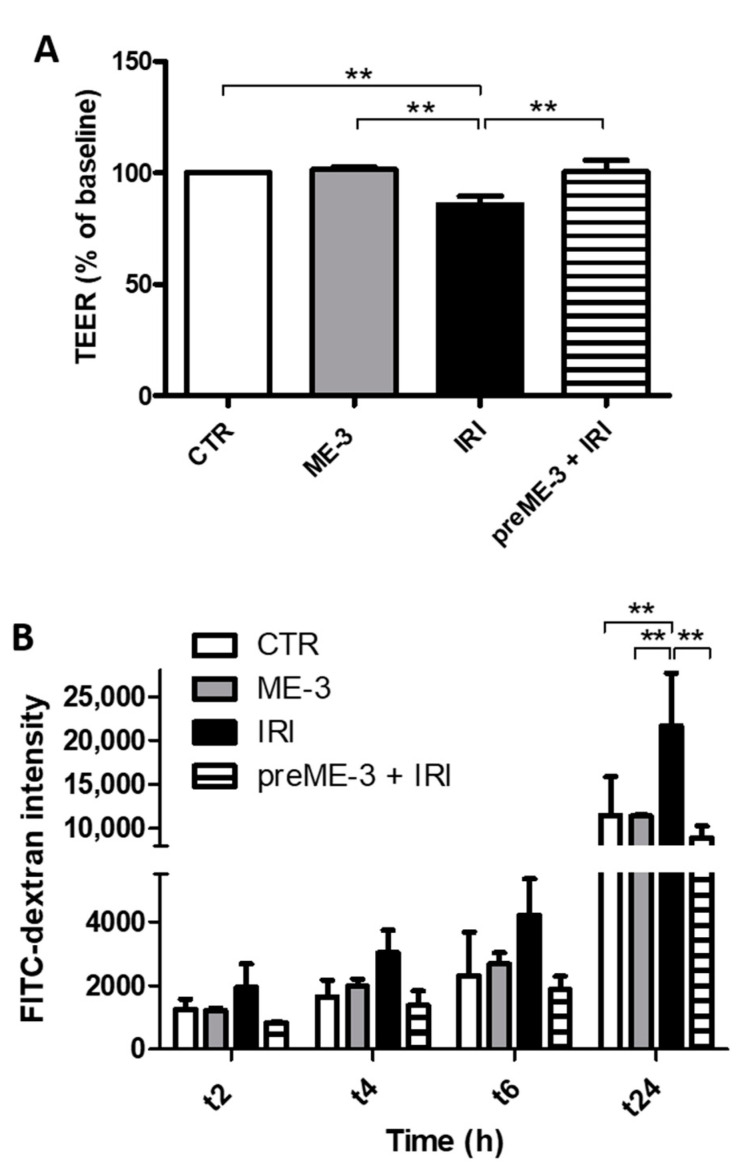
Effect of ME-3^®^ on the permeability of Caco-2 monolayers subjected to Irinotecan treatment. Permeability evaluated in Caco-2 monolayer cells treated for 24 h with ME-3^®^ (10^7^ CFU/mL) and IRI (155 µg/mL) alone or in combination (preME-3^®^ + IRI). (**A**) TEER values were expressed as TEER % of baseline (untreated control), quantified by measuring the differences between the TEER values at 24 h; values were expressed as means ± SDs. (**B**) Time course of dextran flux across Caco-2 monolayer cells. (FITC)-4kDa dextran was added to the apical chamber, and the fluorescence intensity of the medium in the basolateral chamber was measured at different time points after treatment. Three independent transwells were tested for each condition, and values were expressed as means ± SDs. Asterisks represent significant differences between indicated groups: ** *p* < 0.01.

**Figure 3 ijms-24-06225-f003:**
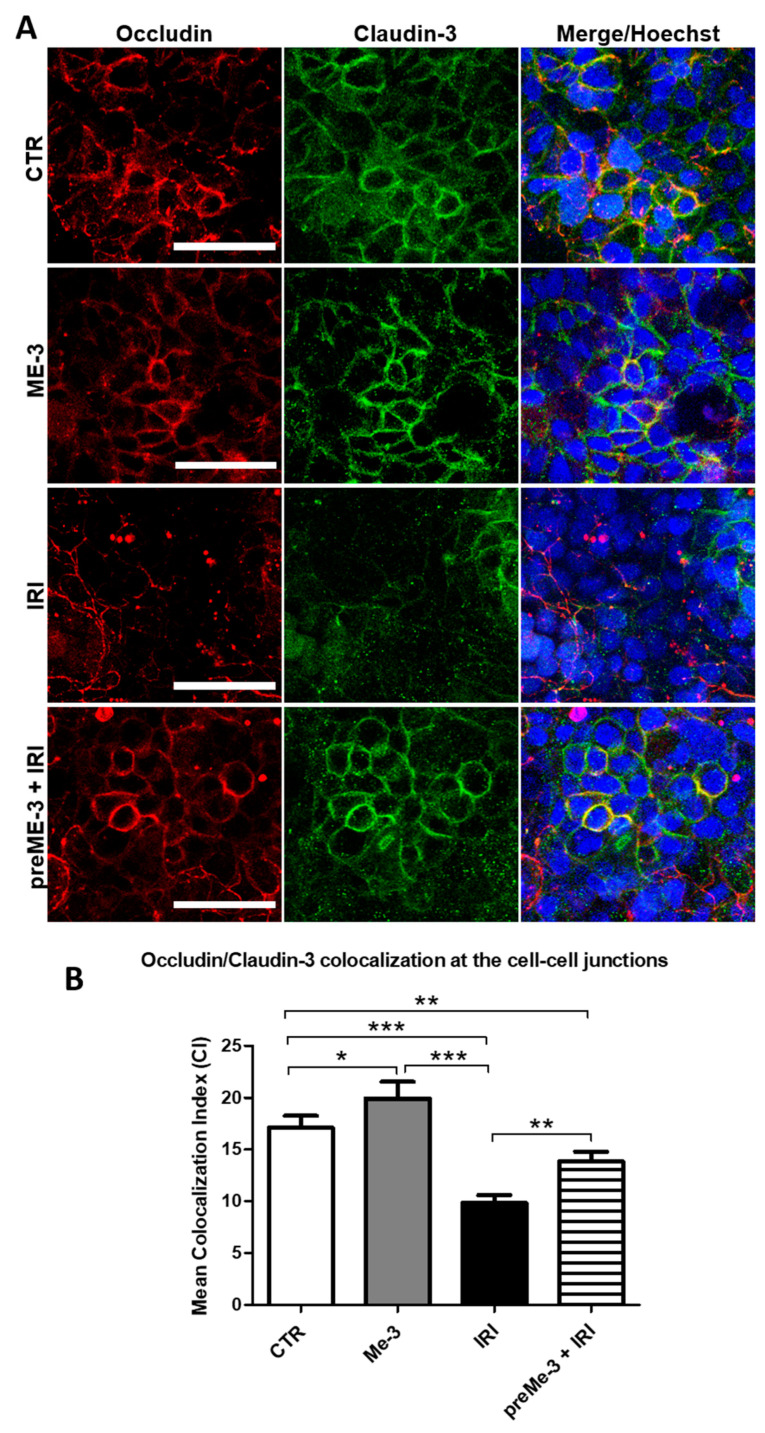
Effects of *L. fermentum* ME-3^®^ on the expression and localization of proteins involved in tight junction (TJ) functionality. (**A**) Confocal microscopy showing the double immunofluorescence staining of Occludin (red hue) and Claudin-3 (green hue), performed as described in the MM section, in controls (untreated and ME-3^®^ treated cells) and IRI-treated cultures grown in the presence or absence of ME-3^®^ pretreatment. The colocalization at the cell–cell contact of the two TJ components is represented by the yellow hue. Cell nuclei were counterstained with 0.2% Hoechst. Bars = 50 μm. The bar graph in (**B**) reports the mean Occludin/Claudin-3 colocalization indexes (CIs), calculated as described in the Materials and Methods, for controls and treated cells. Quantification of CIs was carried out by analyzing a minimum of 100 cells/sample, and the results are the means ± SDs from three independent experiments (*n* = 3). Significance (one-way ANOVA + Tukey’s multiple comparison test): * *p* < 0.05, ** *p* < 0.01, *** *p* < 0.001.

**Figure 4 ijms-24-06225-f004:**
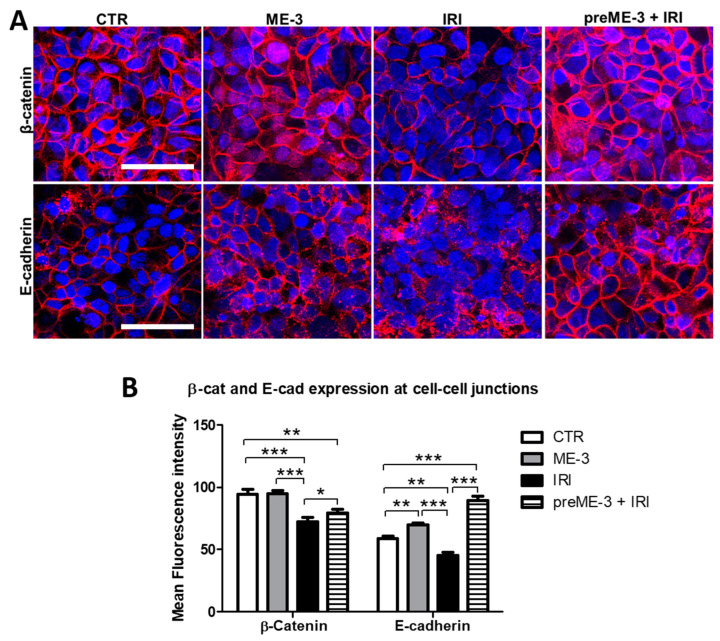
Effects of *L. fermentum* ME-3^®^ on the expression and localization of proteins controlling the adherens junctions (AJs). (**A**) Confocal microscopy showing the immunofluorescence staining of β-catenin and E-cadherin at the AJs, in controls (untreated and ME-3^®^ treated cells) and in IRI-treated cultures grown in the presence or absence of ME-3^®^ pretreatment. Cell nuclei were counterstained with 0.2% Hoechst. Bars = 50 μm. The bar graph in (**B**) represents the quantitative analysis of the distribution of the two AJ components, in controls and treated cells, measured as described in the Materials and Methods. Quantification of fluorescence intensity was carried out by analyzing a minimum of 100 cells/sample, and the results are the means ± SDs from three independent experiments (*n* = 3). Significance (one-way ANOVA + Tukey’s multiple comparison test): ** p* < 0.05, ** *p* < 0.01, *** *p* < 0.001.

**Figure 5 ijms-24-06225-f005:**
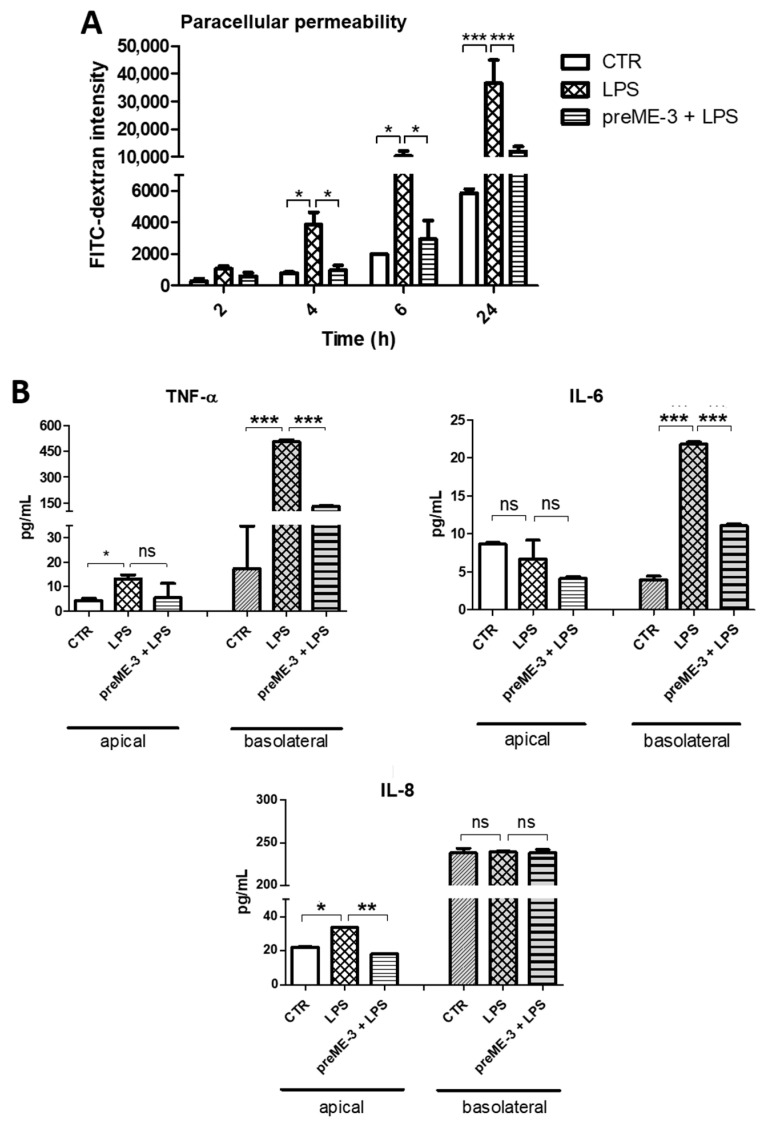
Effects of *L. fermentum* ME-3^®^ on permeability and cytokine production in Caco-2/U937 co-cultures. (**A**) Time course of dextran flux across Caco-2 monolayer cells pretreated or not with ME-3^®^ (10^7^ CFU/mL) in LPS-inflamed Caco-2/U937 co-cultures. (FITC)-4kDa dextran was added to the apical chamber, and the fluorescence intensity of the medium in the basolateral chamber was measured at different time points. (**B**) Pro-inflammatory cytokines TNFα (right), IL-6 (middle), and IL-8 (left) in the apical or basolateral medium collected from the inflamed Caco-2/U937 co-culture chamber (see Materials and Methods). Values are expressed as the means of three separate experiments ± SDs: * *p* < 0.05, ** *p* < 0.01, *** *p* < 0.001; ns = not significant.

**Figure 6 ijms-24-06225-f006:**
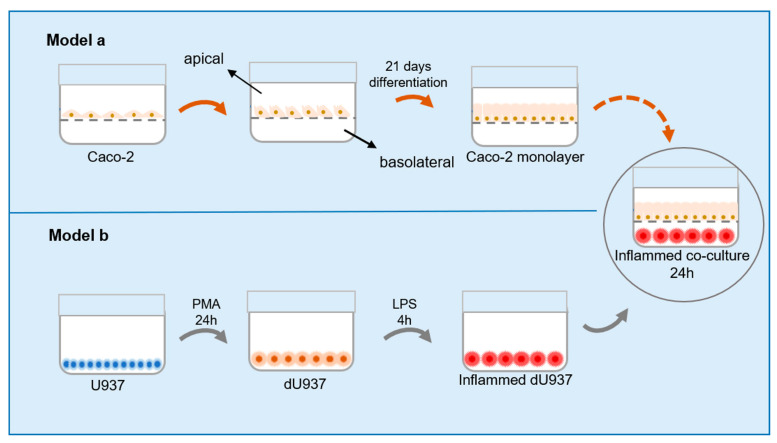
Schematic representation of intestinal barrier and intestinal inflammation models used.

## Data Availability

All the data produced in this study are reported in the article. The primary data files are available from the corresponding author upon reasonable request.

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
