# Peer review of "Protective Effect of Limosilactobacillus fermentum ME-3 against the Increase in Paracellular Permeability Induced by Chemotherapy or Inflammatory Conditions in Caco-2 Cell Models"

_ijms, 2023, doi:10.3390/ijms24076225_

Round 1

Reviewer 1 Report

This seems to be an important study, worthy of publication.  I believe its findings are novel, but I'm not sure to what extent. That is, I would like to know if anyone has found if "generic" probiotics available widely, like Lactobacillus fermentum, could work just as good? In other words, I'd like the authors to tell us if/why they are making the proprietary LF ME-3 seem superior, when in fact they don't compare it to anything else, and we don't know if they do/or don't know that it is? 

The study provides in vitro evidence that a proprietary probiotic, ME-3, meets model criteria to become a helpful adjunctive with chemotherapeutic, IRI, for treating colon cancer. The criteria are nice: (1) ME-3 does not show worrisome effects alone on the model human cell lines Caco-2 and U937.  It doesn’t harm the cells and the only response I see alone is a “useful” small rise in trans electrical resistance (TEER) ; (2) In pretreatment combination, ME-3 does not interfere with the desirous anti-cancer effects of IRI.  This means the treatment of IRI still works.  (3) But, ME-3 pretreatment prevents an unwanted side-effect of IRI treatment which was a leaky gut model state. This was demonstrated in a variety of ways down to certain molecular rearrangements of surface markers on Caco-2 epithelial cells.  By showing similar anti-leaky value when ME-3 is given before application of a very different type of leaky gut causation (the noxious bacterial toxin, LPS), the science could go further and also say that by whatever mechanism ME-3 acts with IRI, it is unlikely to involve directly modifying or degrading the IRI drug itself.   The study is fine up to that point. 

However, when the study comes to the cytokines, I question “what does this data tell us about the oncotherapy adjuvant role of ME-3”. That is by showing that the anti-leak utility of ME-3 in combination with LPS is anti-inflammation, i.e., by lowering the U937 macrophage pro-inflammatory cytokine release.  The first issue I have here is that these scientists fail to present any data whatsoever about the same kind of cytokines following IRI treatment either alone or in combo with ME-3.  The second problem is that setting-up these studies brought forward the macrophage cell line which wasn’t ever employed in the ME-3 + IRI studies (although the abstract makes one think it was, BTW, in my opinion).  Thirdly, one wonders exactly what the source of the cytokine rise was.  For comparison, were there any cytokines coming from the Caco-2 cultures alone?  What about the cytokine levels in the 10% FBS?  Thus, the current findings with cytokines – although very intriguing about LPS reversal of leaks – seem one or two experiments short of convincing me of the mechanism of action of ME-3 during IRI chemotherapy (which for me is the main thrust of this paper).  [But I do hope the cytokine data can someday somewhere appear in publication because it is intriguing on its own regarding leaky guts!]

I also would ask the scientists to comment in the paper a bit about the following:

1.      If we look at MTT and trypan blue staining data of the Caco2 cells in response to IRI alone we see big effects (Fig. 1 down 50% in MTT and down 70% in trypan blue), but not so much with the real leaky gut measures. So, could you comment on why these discrepancies compared to:

·         TEER was down only 15% (Fig. 2A).  

·         Occludin/cloudin tight junction markers down about 45% (Fig. 3).

·         20% drop in beta-cantenin (Fig. 4).

·         There was no drop in E-cadherin due to IRI alone vs. control untreated (Fig. 4).

Does the co-culture with U937 cells, alone, alter the tight junction measures?

Overall, it seems like these scientists did good work, so I feel conflicted calling for a major revision.  I guess it needs to answer my concerns somewhere between a major revision and a minor revision. 

Author Response

REVIEWER #1: This seems to be an important study, worthy of publication.  I believe its findings are novel, but I'm not sure to what extent. That is, I would like to know if anyone has found if "generic" probiotics available widely, like Lactobacillus fermentum, could work just as good? In other words, I'd like the authors to tell us if/why they are making the proprietary LF ME-3 seem superior, when in fact they don't compare it to anything else, and we don't know if they do/or don't know that it is?

ANSWER: We thank the reviewer for raising this critical point, thus allowing us to avoid misunderstanding. In this study, we focused on the protective action of L. fermentum ME-3 on intestinal cell permeability without making a comparison with other probiotics. This implies the impossibility of indicating ME-3 as superior, and in fact, we have yet to affirm in the text that L. fermentum ME-3 is the more suitable probiotic for facing leaky gut syndrome. The point is that our work was purposely concentrated on ME-3 only, a probiotic largely employed in clinical practice and proved to be effective and safe. Our investigation specifically aimed to demonstrate its ability to prevent the alteration in cell permeability caused by harmful stimuli, such as chemotherapy or inflammation, hypothesizing a mode of action.

REVIEWER #1: The study provides in vitro evidence that a proprietary probiotic, ME-3, meets model criteria to become a helpful adjunctive with chemotherapeutic, IRI, for treating colon cancer. The criteria are nice: (1) ME-3 does not show worrisome effects alone on the model human cell lines Caco-2 and U937.  It doesn’t harm the cells and the only response I see alone is a “useful” small rise in trans electrical resistance (TEER) ; (2) In pretreatment combination, ME-3 does not interfere with the desirous anti-cancer effects of IRI.  This means the treatment of IRI still works.  (3) But, ME-3 pretreatment prevents an unwanted side-effect of IRI treatment which was a leaky gut model state. This was demonstrated in a variety of ways down to certain molecular rearrangements of surface markers on Caco-2 epithelial cells.  By showing similar anti-leaky value when ME-3 is given before application of a very different type of leaky gut causation (the noxious bacterial toxin, LPS), the science could go further and also say that by whatever mechanism ME-3 acts with IRI, it is unlikely to involve directly modifying or degrading the IRI drug itself.  The study is fine up to that point.

OUR COMMENT: We would like to thank the reviewer for appreciating the criteria used in the first part of our work and for understanding the value of the results obtained.

REVIEWER #1: However, when the study comes to the cytokines, I question “what does this data tell us about the oncotherapy adjuvant role of ME-3”. That is by showing that the anti-leak utility of ME-3 in combination with LPS is anti-inflammation, i.e., by lowering the U937 macrophage pro-inflammatory cytokine release.  The first issue I have here is that these scientists fail to present any data whatsoever about the same kind of cytokines following IRI treatment either alone or in combo with ME-3.  The second problem is that setting-up these studies brought forward the macrophage cell line which wasn’t ever employed in the ME-3 + IRI studies (although the abstract makes one think it was, BTW, in my opinion).  Thirdly, one wonders exactly what the source of the cytokine rise was.  For comparison, were there any cytokines coming from the Caco-2 cultures alone?  What about the cytokine levels in the 10% FBS?  Thus, the current findings with cytokines – although very intriguing about LPS reversal of leaks – seem one or two experiments short of convincing me of the mechanism of action of ME-3 during IRI chemotherapy (which for me is the main thrust of this paper).  [But I do hope the cytokine data can someday somewhere appear in publication because it is intriguing on its own regarding leaky guts!]

ANSWER: Undoubtedly, the idea of analyzing the effect of ME-3 on the alteration of intestinal barrier permeability induced by IRI, as well as on the release of inflammatory cytokines, also in the co-culture model (Caco2 cells and macrophages), is very attractive. But, as stated in the title and better specified in the abstract of the revised version, in this work, the aim has been to evaluate the beneficial effect of ME-3 on the permeability of the intestinal barrier in two pathological conditions, the one deriving from chemotherapeutic treatment and the one occurring in intestinal inflammation (experimentally induced by LPS), pathological conditions which may coexist and interact but are not necessarily so. Obviously, the results obtained are very stimulating for further studies aimed at assessing the protective effect of ME-3 in the IRI-induced inflammation in the co-culture model, also eventually deepening some mechanistic aspects.

REVIEWER #1: I also would ask the scientists to comment in the paper a bit about the following:

  1.      If we look at MTT and trypan blue staining data of the Caco2 cells in response to IRI alone we see big effects (Fig. 1 down 50% in MTT and down 70% in trypan blue), but not so much with the real leaky gut measures. So, could you comment on why these discrepancies compared to:
  • TEER was down only 15% (Fig. 2A).
  • Occludin/claudin tight junction markers down about 45% (Fig. 3).
  • 20% drop in beta-cantenin (Fig. 4).
  • There was no drop in E-cadherin due to IRI alone vs. control untreated (Fig. 4).

ANSWER: As specified in the MM sections 4.2.-4.4, cytotoxicity and intestinal permeability measurements (the last including immunofluorescence staining of occludin/claudin, β-catenin, and E-cadherin) were done on two different cell density conditions; in particular, the cytotoxicity has been evaluated on cells plated and treated after an overnight adhesion and analyzed after a maximum of 72h, while the permeability tests and immunofluorescence were performed on a barrier model obtained with cells plated, grown for 21 days and then analyzed. It follows that, in this last condition, the cells were not only more numerous but exhibited tighter junctions at the cell-cell contacts, and from this possibly derives the “discrepancy” of measures evidenced by the referee. However, the results obtained in all the tests (which are performed with different methods) show similar trends of effects and are statistically significant. As it concerns the “no drop in E-cadherin due to IRI alone vs control untreated” evidenced by the reviewer, actually, the data reported in the bar graph of Fig. 4B show that this “drop” has been recorded, but it was not significant. In this regard, we are very grateful to the referee for this comment since it allows us to implement the results on E-cadherin expression with additional measures made in ongoing experiments and to perform a more appropriate statistical analysis (One-way ANOVA + Tukey multiple comparison test, instead of t-Test), which increased the significance of the results on E-cadherin expression at the cell-cell junctions. We revised Figure 4B consequently.

REVIEWER #1: Does the co-culture with U937 cells, alone, alter the tight junction measures?

ANSWER: The co-culture of Caco-2 with dU937 (Model b) does not alter the cell-cell contacts, as demonstrated by the quite similar measures of dextran flux across the Caco-2 monolayer cells performed at T0 on both models a and b (FITC-dextran intensity at T0 equal to 2113 and 2141.5 in Model a and b, respectively). This information has been added in the Results section (lines 246-249)

REVIEWER #1: Overall, it seems like these scientists did good work, so I feel conflicted calling for a major revision.  I guess it needs to answer my concerns somewhere between a major revision and a minor revision. 

Reviewer 2 Report

This MS evaluated the effect of Lactobacillus fermentum ME-3 (L. fermentum ME-3) on the extracellular permeability in irinotecan-induced and in co-cultured cells LPS-induced Inflammation. Experimental results are clear. However, it is very disconcerting that this result obtained in cell experiments is described as if it could be applied directly to humans. Therefore, the authors should take a hard look at the results obtained.

Comments

(1) L103: Since this test is an MTT assay, it indicates cell viability. Therefore, the term "growth" is inappropriate.

(2) L118-119: It is stated that L. fermentum ME-3 has been shown to inhibit the pro-apoptotic effects of IRI, but no results are presented to support such.

(3) Fig.2A: Why is there no error bar at t0. I think the average value of n3 is 100, and it should represent the variation of n3.

(4) Fig.3B: The authors use t-tests, but repeated use of t-tests is inappropriate. Please change to an appropriate method.

(5) L191: In the previous test (Fig. 2), the reaction was allowed to run for 24 hours; this test is for 48 hours. If there is any reason for this, it should be stated.

(6) Fig.4B: The authors use t-tests, but repeated use of t-tests is inappropriate. Please change to an appropriate method.

(7) L234: Only in this study, there is no L. fermentum ME-3 alone added group. Is there any reason for this?

(8) Discussion: Please provide a hypothesis as to by what mechanism the barrier function was improved by adding L. fermentum ME-3, and provide references.

(9) Limitation: Limitation should be established. The authors have developed their logic to apply the results to humans. However, I do not believe the results can be easily applied to humans. In fact, in the case of humans, there are many intestinal bacteria other than lactobacilli living in the intestinal tract, so the ingestion of this bacteria may not be effective.

(10) Section 4.4: IRI is a drug used intravenously. Therefore, it is unlikely that IRI is present in the intestinal tract in vivo. Please indicate why the authors added IRI to the apical chamber as well.

(11) Section 4.4: Similar to comment 10. When L. fermentum ME-3 is applied to humans, it should be present on the intestinal side. Why do the authors add basolateral to the basolateral chamber as well? I believe that experimental conditions do not mimic a human.

Author Response

REVIEWER #2: This MS evaluated the effect of Lactobacillus fermentum ME-3 (L. fermentum ME-3) on the extracellular permeability in irinotecan-induced and in co-cultured cells LPS-induced Inflammation. Experimental results are clear. However, it is very disconcerting that this result obtained in cell experiments is described as if it could be applied directly to humans. Therefore, the authors should take a hard look at the results obtained.

Comments

REVIEWER #2: (1) L103: Since this test is an MTT assay, it indicates cell viability. Therefore, the term "growth" is inappropriate.

ANSWER: We agree with the reviewer, and replace the term “growth” with “viability

REVIEWER #2: (2) L118-119: It is stated that L. fermentum ME-3 has been shown to inhibit the pro-apoptotic effects of IRI, but no results are presented to support such.

ANSWER: Our statement has been made based on the measurement of the number of cells exhibiting apoptotic nuclei (Figure 1C) and by taking into account that IRI, functioning as an inhibitor of topoisomerase1, is known to induce apoptosis. In any case, since we did not use any other specific tests to evaluate the apoptosis induction by IRI, we replaced the term “pro-apoptotic” with the word “cytotoxic” 

REVIEWER #2: (3) Fig.2A: Why is there no error bar at t0. I think the average value of n3 is 100, and it should represent the variation of n3.

ANSWER: Perhaps, there may have been a misunderstanding since the value 100 showed at T0 in Fig.2A is not an average value of n3, but it has been arbitrarily fixed as a baseline value of untreated cells to which correlate in percent the TEER values recorded in treated samples at T24h. In any case, since the bar graph at T0 could be misleading for the reader, we removed it from Fig. 2A, leaving only the data recorded at T24. We thank the referee for the comment that allowed us to improve the clarity of Fig. 2A

REVIEWER #2:  (4) Fig.3B: The authors use t-tests, but repeated use of t-tests is inappropriate. Please change to an appropriate method.

ANSWER: We are very grateful to the referee for this comment and for that reported at the following point (5) since they allow us to implement the results on E-cadherin expression with additional measures made in ongoing experiments and to perform more appropriate statistical analysis (One-way ANOVA + Tukey multiple comparison test, instead of t-Test) for Occludin/Claudin colocalization index and for β-catenin and E-Cadherin expression at the cell-cell junctions, which increased the significance of the results and improve their value.

REVIEWER #2: (5) L191: In the previous test (Fig. 2), the reaction was allowed to run for 24 hours; this test is for 48 hours. If there is any reason for this, it should be stated.

ANSWER: We thank the referee for the observation: it had been a typing error. The correct time of treatment was 24h. We made the correction in the text (line 197)

REVIEWER #2: (6) Fig.4B: The authors use t-tests, but repeated use of t-tests is inappropriate. Please change to an appropriate method.

ANSWER: see the answer to the referee’s comment at point (4).

REVIEWER #2: (7) L234: Only in this study, there is no L. fermentum ME-3 alone added group. Is there any reason for this?

ANSWER: In the experiments performed on the in-vitro model of intestinal inflammation, we took into account the results obtained from the previous experiments in which we recorded no negative effects induced by ME-3 treatment alone on the intestinal barrier. For this reason, we did exclude this group from these sets of experiments.

REVIEWER #2: (8) Discussion: Please provide a hypothesis as to by what mechanism the barrier function was improved by adding L. fermentum ME-3, and provide references.

ANSWER: In this work, we did not deepen the mechanistic aspects of the protective effects of ME-3 on the intestinal barrier but only explored and described these effects. Thus, we have no experimental data on this issue, but can only make some hypotheses, also based on data reported in the literature. In our view, the primary mechanism L. fermentum ME-3 uses to maintain intestinal integrity is the upregulated expression and distribution of proteins belonging to the apical junctional complex, as already reported for other probiotics (Ref. #46 of the revised version). The reinforcement of the mucosal barrier can halt the damage caused by Irinotecan and the LPS-induced inflammation. In addition, the anti-inflammatory properties exhibited by ME-3 that prevent the secretion of pro-inflammatory cytokines are also a mechanism to enhance barrier function. Furthermore, an additional hypothesis is that the strong antioxidative properties of L. fermentum Me-3 (18) permit a protective effect against Irinotecan by reverting the oxidative imbalance evoked by the chemotherapic. A similar mechanism was observed in CaCo-2 cells incubated with Irinotecan and flavonoid luteolin, which can attenuate irinotecan-induced oxidative stress by its scavenging property. (Ref. #47 of the revised version). We added these hypotheses on the mechanism of action of ME-3 in the discussion (lines 361-374)

REVIEWER #2: (9) Limitation: Limitation should be established. The authors have developed their logic to apply the results to humans. However, I do not believe the results can be easily applied to humans. In fact, in the case of humans, there are many intestinal bacteria other than lactobacilli living in the intestinal tract, so the ingestion of these bacteria may not be effective.

ANSWER: We understand the referee's concerns, but we actually provide results obtained in two cellular models validated and often used for testing the efficacy of different synthetic or nature-derived molecules on the intestinal barrier. Obviously, our work possesses the intrinsic limitation of an in-vitro study, common to all preclinical in vitro studies, for which the results obtained should be subsequently validated by later studies on animals and/or human subjects. We mentioned this limitation in the discussion (lines 375-376)

REVIEWER #2: (10) Section 4.4: IRI is a drug used intravenously. Therefore, it is unlikely that IRI is present in the intestinal tract in vivo. Please indicate why the authors added IRI to the apical chamber as well.

ANSWER: We are afraid we didn't get what exactly the referee's criticism is. Although administrated intravenously, Irinotecan has been shown to contribute to intestinal permeability alteration (Ref.14 of the revised version). Therefore, intending to verify an effect on the epithelial component of the intestinal barrier model, we added Irinotecan to the apical chamber to directly treat the Caco2 cells, as it is usually done for an in-vitro experiment, and also based on a published paper performed on Caco2 cells treated with Irinotecan. (ref #47 of the revised version).

REVIEWER #2: (11) Section 4.4: Similar to comment 10. When L. fermentum ME-3 is applied to humans, it should be present on the intestinal side. Why do the authors add basolateral to the basolateral chamber as well? I believe that experimental conditions do not mimic a human.

ANSWER: Since it is known that in inflammatory conditions (as well as in intestinal cancer), there may be present lesions of the barrier that could be enriched of macrophagic infiltrates, we cannot exclude that there may be a beneficial effect even directly on the macrophagic cells and, for this reason, we added ME-3 also to the basolateral chamber.

Round 2

Reviewer 1 Report

Thank you for taking my comments to heart. Good luck with future studies.

Reviewer 2 Report

I am satisfied with the revisions that have been made by the authors.